# Psychological Consequences of Fear of COVID-19: Symptom Analysis of Triggered Anxiety and Depression Disorders in Adolescents and Young Adults

**DOI:** 10.3390/ijerph192114171

**Published:** 2022-10-29

**Authors:** Isabel Mercader Rubio, Pilar Sánchez-López, Nieves Gutiérrez Ángel, Nieves Fátima Oropesa Ruiz

**Affiliations:** Department of Psychology, Universidad of Almería, 04120 Almería, Spain

**Keywords:** COVID-19, fear, anxiety, depression, adolescents, young adult

## Abstract

Among the innumerable consequences of the pandemic caused by COVID-19 are those of a psychological nature, among which we find fear. For this reason, it is important to carry out research on the fear of contagion and its behavior, especially in the population as a whole, and the consequences that these facts entail. The present study examines the fear of contagion and illness by COVID-19 and its relationship with symptoms of anxiety disorders and depression in a total of 1370 participants aged 16 to 29 years. The results indicate that fear of COVID-19, fear of death from COVID-19 infection, and concern that family members and/or friends will be infected with COVID-19 are predictors of symptoms related to depressive disorder. elderly and social phobia.

## 1. Introduction

SARS-CoV-2 (COVID-19) has become a global public health threat [1] due to high infection rates, mass quarantines and increasing numbers of deaths worldwide [2,3]. This pandemic has generated a high level of distress in the general population, manifesting a large increase in symptoms of depression and anxiety [4] but especially in vulnerable populations there has been an increased risk of mental disorders [5], suicide attempts [6] and has increased stigma and discrimination when in contact with people whose symptoms could be infected, causing fear and certain panic behaviors that can have direct negative consequences for disease control [7]. Fear reactions to infectious diseases are considered normal, but the increasing mortality from this virus highlights the importance of conducting research on fear of infection and behaviors, especially in the general community [8].

Surveys in several countries therefore indicate a strong increase in fear and concerns related to this virus [9]. In Belgium of 44,000 participants in April 2020, 20% reported anxiety and 16% reported a depressive disorder, increasing substantially compared with a survey conducted in the same country in 2018 [10]. As fear may be a central construct in explaining these negative individual and societal consequences of the pandemic, it is important to better understand what exactly people fear and to establish relevant predictors [11].

Fear serves to cope with a potential threat. It is an adaptive emotion that if not properly calibrated to the actual threat, can be maladaptive. This fear when excessive can have detrimental effects both at the individual level (phobia and anxiety) and at the social level, panic buying, germ aversion or xenophobia [12]. Furthermore, it triggers safety behaviors such as handwashing, which can decrease virus transmission, but paradoxically can also increase fear and health anxiety [13]. Fear of contracting the virus, fear of lack of basic supplies (food, water, clothing, shelter) have been shown to cause feelings of frustration and distress Bandera-Pastor et al. [14] that tend to be increase by lack of access to adequate information, in particular unclear or contradictory guidelines about the behavior required during confinement and confusion about its purpose Quezada-Scholz [15]. This may be in addition to an already known fear of acquiring associated infections in healthcare settings [16]. Coupled with the possibility that our health or that of our loved ones is affected [17].

Currently, according to [18] emotions such as anxiety, fear, sadness, anger or impatience will be very common in most people. By feeling these emotions, they share the function of preserving life and keeping us alert to what is threatening us, as in this case of COVID-19. Fear and anxiety help orient us towards the threat or harm Gómez-Beceera et al. [19]. According to research, people’s fears about the coronavirus are related to different content. Specifically, Ref. [18] developed and validated the COVID-19 stress scale (CSS) and identified coronavirus-related stressors and anxiety symptoms in two large samples from Canada and the United States.

Among the groups identified as being most vulnerable are, among others, children and young people. Fears of infecting oneself and others have been identified as specific to the current pandemic in this age group [20], and a positive relationship has also been established between fear of COVID-19 and emotional problems in childhood and adolescence from the time of confinement onwards [21,22]. Adolescents and young adults appear to show higher levels of anxiety, depression and stress, so it would be expected that the young adult population, aged approximately 18–30 years, has been affected to a greater extent than younger children and older adults [23,24,25].

It is precisely these theoretical antecedents that justify our investigation, in order to verify if such consequences have occurred in our closest environment, if the same occurs in those certain evolutionary stages, and if we can conclude that such consequences have occurred to general level in the sample population of our study.

This study aims to analyze the symptoms of anxiety and depressive disorders and their relationship with fear of COVID-19 in adolescents and young adults.

The hypotheses established in this study are as follows:

**Hypothesis** **(H1).***There is a direct and positive relationship between fear of COVID-19 and anxiety or social phobia*.

**Hypothesis** **(H2).***There is a direct and positive relationship between fear of COVID-19 and generalized anxiety disorder*.

**Hypothesis** **(H3).***There is a direct and positive relationship between fear of COVID-19 and depression disorder*.

## 2. Materials and Methods

The method used was correlational, corresponding to an ex post facto design. In it, the data are analyzed retrospectively and comparatively, because the dimensions of the symptoms of anxiety and depression disorders are compared with other types of variables. The dependent variables, specifically, the fear of COVID-19, correspond to the fear of death from having been infected with COVID-19, and the concern that family and/or friends may be infected with COVID-19.

### 2.1. Participants

The sample consists of a total of 1370 participants through simple random sampling. The age range was between 16 and 29 years, of which 62.7% (*n* = 859) were female and 36.9% (*n* = 505) were male. The mean age of the males was 19.78 years (SD = 2.36), while the mean age of the females was 19.68 years (SD = 2.70).

Table 1 presents the sociodemographic data of the sample.

### 2.2. Instruments

The instruments used in this study are as follows:

The Anxiety and Depressive Disorders Symptoms Scale (ESTAD) [26]. It corresponds to a self-report measure whose purpose is to measure the symptoms of different disorders related to both anxiety and stress through a total of 35 items divided into 7 subscales based on a Likert scale (1–4). The subscales pertaining to generalized anxiety disorder, social phobia and major depressive disorder were chosen for this study. The psychometric properties of this instrument show adequate evidence of convergent and discriminant validity: generalized.

Anxiety disorder α = 0.86; social phobia α = 0.92; and major depressive disorder α = 0.87; obtaining an overall scale score of α = 0.96 [27]. For this paper, generalized anxiety disorder α = 0.86; social phobia α = 0.77; and major depressive disorder α = 0.80; obtaining an overall scale score of α = 0.93.

The Fear of COVID-19 Scale-FCV-19S [28]. This is a scale composed of 7 factors that assesses the fear of being infected by COVID-19 using a Likert scale (1–5) [29]. Only the factors related to “I am very afraid of COVID-19”, “I am afraid of losing my life due to COVID-19”, and “I am worried that my family and/or friends will catch COVID-19” were chosen for this study. The psychometric properties of this instrument show adequate evidence of convergent and discriminant validity, as it scored α = 0.88 [30] and α = 0.86 [31]. For this study, a score of α = 0.82 was obtained.

### 2.3. Data Analysis

Data analysis corresponds to descriptive statistics (mean, standard deviation and bivariate correlations), reliability analysis and structural equation modelling (SEM) to differentiate the relationships determined in the hypothesized model. Specifically, we employed Joreskog’s practice for the analysis of the covariance structure to a Multiple Indicator-Multiple Causes (MIMIC) [32,33].

For the acceptance or rejection of the proposed model we take as a basis the scores obtained in the fit indices Hu et al. [34]: TLI (Tucker–Lewis index), SRMR (standardized root mean square residual) and RMSEA (root mean square error of approximation). Thus, the appropriate indices are: TLI value above 0.95; SRMR values below 0.06; and RMSEA values below 0.08. These analyses were performed with SPSS version 26 and R statistical analysis software.

## 3. Results

### 3.1. Relationship between Psychological Consequences and Fear of COVID-19

The relationship between generalized anxiety, social phobia and major depressive disorder, and fear of COVID-19 itself, fear of life due to COVID-19, and worry that family members and/or friends will catch COVID-19 were analyzed. As can be seen in Table 2, the correlations between the study variables were positive, reflecting reciprocity between the study variables.

### 3.2. Structural Equation Mode

The hypothetical model of predictive relationships (Figure 1) provides that the fit indices are grouped into three groups:Global or Absolute fit indices: which evaluate the model in general, among which we have Chi-square, RMSEA = 0.03, GFI = 0.99.Incremental or Comparative Fit Indices: these compare the proposed model with the model of independence or absence of relationship between the variables; among these indices are the NNFI = 0.97; TLI= 0.97; CFI = 0.95; IFI = 0.98.Parsimony indices: They evaluate the quality of the model fit according to the number of coefficients estimated to achieve this level of fit, in this category we have the chi-squared ratio between the degrees of freedom (X2/gL), and AGFI = 0.97.

### 3.3. Relationships Established in the Structural Equation Model

The relationships established in the structural equation model are specified below: One’s own fear of COVID-19, fear of losing one’s life due to COVID-19, and worry that family members and/or friends will catch COVID-19 correlate directly and positively with major depressive disorder.

Fear of COVID-19 itself, fear of losing one’s life due to COVID-19, and worry that family members and/or friends will catch COVID-19 correlate directly and positively with social phobia. There is no direct and positive correlation between fear of COVID-19 itself, fear of losing one’s life due to COVID-19, and worry that family members and/or friends will catch COVID-19 and generalized anxiety disorder.

## 4. Discussion and Conclusions

The aim of this research was to analyze the symptoms of anxiety and depressive disorders caused by fear of COVID-19 in adolescents and young adults. The main manifestations of this research lie in the demonstration of the existence that fear of COVID-19, fear of death due to having been infected with COVID-19, and concern that family members and/or friends might be infected with COVID-19 are predictors of symptoms related to major depressive disorder and social phobia.

This research is along the same lines as other studies that have looked at fear and pollution [17], fear of economic consequences [35], fear and xenophobia [36], and fear and traumatic stress symptoms on COVID-19 [37]. Therefore, it can be concluded that the instrument used in this study (the Spanish version of the FCV-19S scale) is a valid measure to assess fear in the adult population. These results indicate that Hypothesis 1 is not fulfilled. While the data obtained allow us to accept Hypotheses 2 and 3.

Fear of COVID-19 could be a traumatic, stressful experience, and is associated with depression and anxiety; therefore, it may be fortunate to have a validated follow-up measure. For these reasons, early identification of fear level could facilitate interventions to treat disorders such as anxiety and depression in the general population, allowing the development and targeting of strategies to improve fear management [36].

Regarding mental health problems, research by Román et al. [38] identified the presence of higher affective states of depression, anxiety and stress than those found in two pandemic studies in which the same instrument was administered, one in China, Wang et al. [39] and the other in the Basque Autonomous Community [40].

APA [41] has established that the cause of depression is due to chemical changes in the body that negatively influence emotions and thought processes. Depression initially shows a lack of balance in certain mental and emotional aspects of life. Thus, people with depression feel helpless, overwhelmed and blame themselves for having these negative feelings, and some have suicidal thoughts, which has been aggravated by the COVID-19 pandemic [42,43].

People with diagnosed mental health problems or addiction disorders may be especially vulnerable in such an emergency. Mental health conditions (depression, anxiety, bipolar disorder or schizophrenia) affect not only how a person thinks, feels and behaves, but also their ability to relate and function in daily life [36,44,45]. These patients should continue treatment and be vigilant for new or worsening symptoms. Feelings of isolation, depression, anxiety are known to increase the risk of suicide. People are more likely to experience these feelings during a crisis such as the one we have experienced. To prevent suicidal thoughts and behaviors, family and community support, being connected and having access to virtual or face-to-face therapy are essential to counteract suicidal thoughts and behaviors, especially during a crisis such as the COVID-19 pandemic [22,28].

Our findings confirm evidence from the literature that adolescents and young adults appear to show significant vulnerability to fear of COVID-19 and associated mental health problems. This reinforces the need to support this sector of the population, especially those who show higher levels of fear, in order to establish public policies that can help overcome the negative effects of the pandemic and mitigate its consequences in the medium and long term.

One of the limitations of this study corresponds to the sample size, so we must be cautious with the results obtained. Therefore, future lines of research will aim to increase the sample size, as well as to take into account different psychological variables to be considered as determinants of fear, such as levels of emotional intelligence, engagement, or self-esteem.

The main conclusion reached by this manuscript is that, as the existing specific literature has already shown, COVID-19 has had different negative psychological consequences among certain vulnerable groups, including adolescents and young adults. The results presented allow us to verify, within these psychological consequences, how the symptoms of generalized anxiety or major depression disorder are a consequence of the fear of COVID-19.

## Figures and Tables

**Figure 1 ijerph-19-14171-f001:**
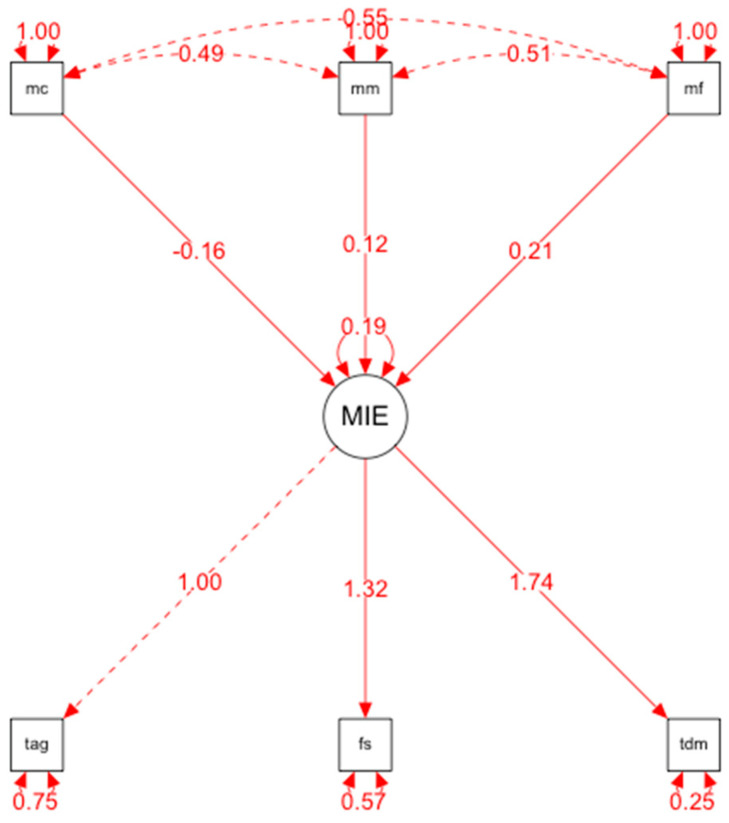
Structural equational modelling.

**Table 1 ijerph-19-14171-t001:** Sample.

	Under 22 Years	Over 22 Years	Total	Infected	No Infected
Female	656 (86.54%)	102 (13.45%)	758	56 (4.7%)	674 (57%)
Male	438 (86.73%)	67 (13.26%)	505	419 (35.4%)	28 (2.4%)
Total	1094	169		475	702

**Table 2 ijerph-19-14171-t002:** Preliminary analyses.

	1	2	3	4	5	6
1. Generalized anxiety disorder		0.518 **	0.554 **	0.302 **	0.263 **	0.258 **
2. Social phobia			0.490 **	0.252 **	0.218 **	0.159 **
3. Major depressive disorder				0.012	0.056	0.056
4. Fear of COVID-19					0.575 **	0.436 **
5. Fear of dying of COVID-19						0.302 **
6. Family members infected						

Note. ** *p* < 0.01.

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
