# Peer review of "Psychological Consequences of Fear of COVID-19: Symptom Analysis of Triggered Anxiety and Depression Disorders in Adolescents and Young Adults"

_ijerph, 2022, doi:10.3390/ijerph192114171_

Round 1

Reviewer 1 Report

This research paper describes research investigating psychological symptoms triggered by the fear of covid-19. I think the manuscript is poorly written and structured and should undergo a lot of editing before being published. Therefore, I have some comments that should improve readability and understandability of the paper. As the authors are not English native speakers, some sentences are hard to comprehend, and grammatically not correct. Therefore, I suggest letting your paper be grammar-checked by a native speaker before being published. I further think that the paper lacks thoroughness; there is no clear rationale, no hypothesis, and the discussion is too short. The authors test a model, which they did not even propose. Many details about the study are missing, and I think this study is not very new or unique, as there have been many studies about covid and psychological disorders like depression and anxiety. Not only do the authors fail on meeting the APA guidelines, but also do they not state a clear hypothesis nor describe the research gap. Therefore, I would suggest to either majorly revise this manuscript or reject it in its present form. Below, you find my comments on the manuscript, which should be considered when revising it. I think in its present form, this manuscript does not meet the criteria for good scientific work.

Major Points:

1.     There is no clear hypothesis or rationale within the introduction of the paper.

2.     p. 3, Data analysis: It is unclear what kind of model you hypothesized. As I stated, there is no clear rationale or hypothesis concerning the relationship between your constructs. Therefore, to me it is unclear which model you tested with structural equation modelling.

3.     p. 4, paragraph „Structural equation model“: I do not understand what this paragraph should state. You provide a couple of fit indices, but to me as the reader it remains unclear whether they are good or not. In the first place, you name Chi-square for example, but give no number. For RMSEA or GFI you do not explain what these numbers mean.

4.     Discussion: The discussion is way too short. There is no reference to the theoretical background, no comparison to other studies investigating similar hypotheses and no detailed discussion of the results. On p. 5, line 164-166, you state: „There is no direct and positive correlation between fear of COVID itself, fear of losing one's life due to COVID, and worry that family members and/or friends will catch COVID and generalised anxiety disorder.“ You did not discuss this result in proper detail. Why is there no relation of fear symptoms to anxiety disorder? This result is very contradictory and should therefore be discussed in more detail. Anxiety and depressive disorder are highly correlated with each other for example.

5.     p. 5, line 190: Why is the sample size a limitation? To me, the sample seems pretty large. Nonetheless, I think it would be good if you could provide a power analysis. This can also be done post-hoc, although I would assume that you did this before collecting and analyzing your data. Otherwise, I do not understand why the sample size would be a limitation.

Minor Points:

1.     p. 1, Introduction: Please name the time period, where SARS-CoV-2 has become a global public threat, describing the starting point and the duration of the pandemic.

2.     p. 1, line 23f: “…and in turn has increased stigma and discrimination when in contact with people whose symptoms could be infected, …”I am not sure what you mean with infected? I assume it should be infectious, meaning that people have symptoms dangerous for others.

3.     p. 1, line 30f: Please rephrase the sentence, it should not start with “See the one conducted…”.

4.     p. 1, line 35: Predictors for what?

5.     p. 1, line 37-39: The end of this sentence does not make sense. Please rewrite.

6.     p. 2, line 49f: “According to 18…” – Do you refer to a study here? Please name the authors before adding the number of the reference to this sentence. The same applies to line 53-54 i.e., “18 developed and validated...”. This is not APA style, and for the reader very difficult to handle. Please change this throughout your manuscript wherever applicable.

7.     p. 2, line 57-60: I do not understand this sentence. Why do you compare those three studies with one another? Did they all investigate depression, anxiety, and stress during the covid pandemic in different countries? Or did the latter two studies investigate a different pandemic? Please clarify, otherwise this sentence makes no sense.

8.     p. 2, line 78f: “Among the groups identified as being most vulnerable are, among others, children 78 and young people.“ Please add one or more references for this statement.

9.     p. 3, line 97: “The sample consists of a total of 1370 participants through simple random sampling.” Why did you apply random sampling? Did you draw your sample from a bigger sample? Please clarify. Please also add information about education levels within your sample if available.

10.  p.3, lines 120-122: “This section may be divided by subheadings. It should provide a concise and precise 120 description of the experimental results, their interpretation, as well as the experimental 121 conclusions that can be drawn.“ Please remove from the manuscript, this is a sentence from the Journal guidelines and has no place in the manuscript.

11.  p. 3, line 119: 3. Results should be removed, this is a clear mistake here.

12.  p. 2, Material and Methods: Where was the study conducted? There is no information about the country, the background of the sample, and whether this was an online survey or part of a bigger project. The latter I would assume, as you stated that you received your sample by random sampling.

13.  p. 3, Data analysis: What kind of reliability analysis did you do? Does this refer to the Cronbach’s Alpha of the applied scales? Please clarify.

14.  p. 4, table 1: Please remove the zero in line 2, column 2 for the value 0,518** - all other values do not have a zero in front. Moreover, it is not APA style to use a comma as a decimal separator. Please use a dot throughout, as we are in English language.

15.  p. 4, line 144: There seems to be a missing letter in modeL.

16.  p. 4, line 145: The word hypothetical is not correct; please use hypothesized here.

17.  p. 6, line 193: I think it should be engagement, not enagement. Please let your whole manuscript be corrected by a native speaker in English.

Author Response

  • I suggest that a native speaker check the grammar of their article before publishing it.

Thank you very much for your contributions, errors of expression and style have been corrected about the language throughout the document to improve its quality by a specialist in translation and interpretation in English language that endorses a level of C1 in English language competence.

  • there is no clear justification,

The justification for this research has been added in the introduction (Líenas 106-109)

  • no hypothesis,

A total of three hypotheses have been added about the manuscript, both in the introduction (lines 108-114), and in the discussion (lines 2012-2013).

  • the discussion is too short.

The discussion has been rewritten (lines 264-284)

  • Not only do the authors fail to meet APA guidelines,

All references have been reviewed according to the journal's citation regulations.

All changes are highlighted in blue

Reviewer 2 Report

My reviews are in the attached file

Author Response

Thank you very much for your contributions:

  • The introduction and discussion has been restructured (lines 264-284)
  • Corrected those sentences that occupied a single paragraph
  • Those phrases proposed for deletion by the reviewer have been deleted
  • References to authors have been reviewed throughout the manuscript
  • Corrected those sentences that were too long
  • Suggested references have been incorporated
  • COVID is used throughout the work

All changes are marked in blue

Reviewer 3 Report

Dear Editor,

I have carefully assessed this manuscript. I think that major editing is needed before this work could be considered for publication.

- the editor should notice that "Institutional Review Board Statement: In demand"

- the title does not provide the type of the study, neither the abstract does

 - when talking about a study a subject (e.g., the first author) should be provided, before reporting the ref. in brackets [20]

- the introduction, despite addressing a recent topic, looks as strongly focused on theoretical messages. Particularly, when dealing with adolescent mental health, the interaction between social media (TikYok, Instagram), mental health (anxiety, depression, eating disorders), and the SARS-COV-2 pandemic should represent a relevant improvement;

- "The [23] has established that the cause of depression is due to chemical changes in 61 the body that negatively influence emotions and thought processes": depression and other mental health conditions are complex phenomena, with no unique explanation. Sentences like this should definitely be canceled and replaced.

- how and where was the enrollment performed?

- "3. Results 119 This section may be divided by subheadings. It should provide a concise and precise 120 description of the experimental results, their interpretation, as well as the experimental 121 conclusions that can be drawn." typo here 

- descriptive statistics for the included sample are absent and need to be reported

- a conclusion after the limitations should be placed

Round 2

Reviewer 1 Report

First of all, I would like to thank the authors for taking my comments seriously and changing their manuscript accordingly. Still, I feel like the paper is extremely difficult to read and understand due to bad english. Therefore, I suggest extensive editing by a native speaker before publishing. In addition, I find the citation style very disturbing, as it completely disrupts every sentence, and thus makes readability super difficult. If the journal is fine with that style of citing, though, I will not complain.

Author Response

Thank you very much for your contributions, errors of expression and style have been corrected about the language throughout the document to improve its quality by a specialist in translation and interpretation in English language that endorses a level of C1 in English language competence.